# Effects of Different Protocols of Moderate-Intensity Intermittent Hypoxic Training on Mental Health and Quality of Life in Brazilian Adults Recovered from COVID-19: The AEROBICOVID Double-Blind Randomized Controlled Study

**DOI:** 10.3390/healthcare11233076

**Published:** 2023-11-30

**Authors:** Eugenio Merellano-Navarro, Marta Camacho-Cardenosa, Gabriel Peinado Costa, Ester Wiggers, Germano Marcolino Putti, Jonatas Evandro Nogueira, Elisangela Aparecida da Silva Lizzi, Átila Alexandre Trapé

**Affiliations:** 1Department of Physical Activity Sciences, Faculty of Education Sciences, Universidad Católica del Maule, Talca 3460000, Chile; emerellano@ucm.cl; 2Clinical Management Unit of Endocrinology and Nutrition—GC17, Maimónides Biomedical Research Institute of Cordoba (IMIBIC), Reina Sofía University Hospital, 14004 Córdoba, Spain; marta.camacho@imibic.org; 3School of Physical Education and Sport of Ribeirao Preto, University of Sao Paulo (USP), Ribeirao Preto 14040900, SP, Brazil; costa.gabriel@usp.br (G.P.C.); germanomputti@usp.br (G.M.P.); jonatasnogueira@usp.br (J.E.N.); 4Ribeirao Preto Medical School, University of Sao Paulo (USP), Ribeirao Preto 14040900, SP, Brazil; wiggers.ester@gmail.com; 5Academic Department of Mathematics, Federal University of Technology—Paraná, Cornélio Procópio 86300000, PR, Brazil; elisangelalizzi@gmail.com

**Keywords:** physical exercise, COVID-19, mental health, quality of life, depression, hypoxic training

## Abstract

The aim of this study was to investigate the effects of different protocols of moderate-intensity intermittent hypoxic training in patients who had recovered from COVID-19 on quality of life (QoL) and mental health. The sample of this clinical trial-controlled double-blind study consisted of 67 participants aged 30–69 years, who were organized randomly according to Normoxia, Hypoxia, Hypoxia Recovery or Control Group. Eight weeks of cycle ergometer training were performed with a frequency of three training sessions per week in normoxic or hypoxic conditions (with or without hypoxic recovery). Health-related QoL and Mental Health Status were evaluated by 12-Item Short Form Survey and Depression Anxiety and Stress Scale instruments, respectively. All training groups improved the QoL’s physical dimensions (Baseline–Post: Normoxia Group 42.1 (11.0)–48.7 (7.0), Hypoxia Group 46.9 (11.8)–53.5 (6.6) and Hypoxia Recovery Group 45.8 (9.2)–51.1 (5.3)) and mental dimensions (Baseline–Post: Normoxia Group 48.8 (7.9)–54.6 (4.6), Hypoxia Group 45.2 (7.7)–53.2 (3.8) and Hypoxia Recovery Group 46.5 (9.7)–52.0 (9.9)). Regarding mental health outcomes, all training groups decreased depressive symptoms (66.7% Normoxia, 31.2% Hypoxia Recovery and 31% Hypoxia groups), anxiety symptoms (46.5% Normoxia, 45.9% Hypoxia Recovery and 39.5% in the Hypoxia groups) and stress symptoms (40.6% Normoxia, 36.3% Hypoxia Recovery and 22.1% Hypoxia groups). Significant statistical difference was not found between groups. Normoxic and hypoxic training showed a similar effect on QoL and the mental health of Brazilian adults who had recovered from COVID-19.

## 1. Introduction

As of January 2023, more than 668 million people were diagnosed with COVID-19, and 6.73 million died from the disease worldwide [1]. In the first moments of the pandemic, scientific concern was aimed at controlling the high number of cases and deaths and developing vaccinations. But later, it was necessary to understand and reverse the consequences of the COVID-19 syndrome [2]. In this context, effective long-term management is now required due to the syndrome’s effects, common disease symptoms and death. In addition, the coronavirus pandemic (COVID-19) presented adverse implications for the mental and physical health of the general population [3]. These implications include anxiety, depression and stress [4]. Post-traumatic stress disorders have been reported as well [5].

In many states of the world, due to the high number of infections and deaths, the continued restrictions, the uncertainty and the indication of social isolation might affect mental health and quality of life (QoL) [6]. In 2017, Brazil led in terms of prevalence of anxiety disorders and ranked fifth in depression rates [7]. A study claims that since the beginning of the encirclement, the number of people with elevated anxiety and depression doubled [8]. One of the main characteristics of post-COVID-19 syndrome is altered functionality, resulting in an inability to perform everyday tasks [9]. It has been reported that a patient’s recovery after an acute outbreak of COVID-19 is associated with the permanence of symptoms and a low QoL. Thus, it is necessary to use strategies for the recovery of the QoL of these patients, reducing the demands on health systems.

In this sense, regular physical activity can help control emotions and even reduce anxiety and the risk of mental disorders. Pre-pandemic studies have shown that a higher level of healthy fitness is associated with better health-related QoL, primarily in the physical component [10]. Different exercise protocols have significantly improved the QoL of adults who have recovered from COVID-19 [11,12]. These improvements could be derived from the improvement in physical capacity and biological and physiological modulation achieved by exercise [9]. In particular, aerobic physical activity has a commonly accepted positive effect on cognitive performance and QoL [13]. Structural changes in the brain and expression of brain neurotransmitters as a hippocampal brain-derived neurotrophic factor could explain these possible effects [14].

However, the disability caused by the disease can make it difficult for these patients to carry out physical exercise programs. In this respect, hypoxic training has been suggested in recent decades as an effective and safe therapeutic strategy. A significant advantage of hypoxic training in the health care setting is that the absolute workload is substantially reduced compared to normoxic training, when the internal load is similar [15]. Furthermore, combined with hypoxic preconditioning, scientific literature has shown additional effects due to mild hypoxia’s protective effect on hippocampal cells’ neural viability [16].

Hypoxic training could become a therapeutic alternative for patients who have recovered from COVID-19 and who present a functional deficit in physical fitness due to persistent symptoms in cardiovascular, hematological and pulmonary systems [17]. In this way, similar physiological responses with less mechanical stress could be observed in hypoxia compared with physical training alone. The scientific literature has previously described the effects of hypoxia training on the mental health and QoL of patients with chronic diseases or cognitive impairments [13,18,19,20]. A previous study combined concurrent training with intermittent hypoxia and showed improvements in the quality of life and mental health of retired, sedentary adults (60–70 years old) after hypoxic training compared with the normoxia group [13]. Later, in a subsequent study [20], these authors showed cognitive improvements in physically active older adults by combining an aerobic exercise program with hypoxic exposure. Authors suggested that a potential brain-derived neurotrophic factor expression could have caused greater QoL due to antidepressant-like effects that induce hypoxia. Combined with hyperoxia, a marked improvement in patients’ subjective perception of quality of life and reduced depression and anxiety were marked in patients with stable angina pectoris. Moreover, these effects were maintained for up to one month after the disease [21]. Similar results were observed in patients diagnosed with coronary artery disease at the end of an eight-week combined hypoxia–hyperoxia exercise exposure and one month after the intervention [18].

Promising preliminary results of hypoxia exposure in animal model studies suggest that these interventions could be of great interest in treating anxiety disorders [22]. However, despite recent studies suggesting intermittent hypoxic preconditioning as a powerful strategy for COVID-19 rehabilitation [23], to our knowledge, there is no research in the scientific literature that has studied the effects of this novel therapeutic strategy on mental health parameters of adults who have recovered from COVID-19. Thus, this study aimed to investigate the effects of different protocols of moderate-intensity intermittent hypoxic training in adults who have recovered from COVID-19 on QoL and the state of their mental health. Based on the previous studies, we hypothesized that exposure to hypoxic training, with a lower mechanical load, would present similar benefits on mental health and QoL to the normoxia group, thus helping the recovery.

## 2. Materials and Methods

The present study is a randomized, double-blind, controlled clinical trial composed of four groups, and was performed from September to December 2020. This was registered after inclusion initiation (Brazilian Clinical Trials Registry, RBR-5d7hkv). More detailed information on the experimental design can be found in the published study protocol [24].

### 2.1. Participants

The sample size is detailed in the published protocol [24], presenting a power of 80% for an n = 84. After the intervention, due to losses, it consisted of 67 participants in the context of a longitudinal study and during the pandemic. By inverse calculation, a power of 76% was obtained [25], which was considered acceptable in the applied scenario of the study. All participants met the following inclusion criteria: (1) being convalescent with COVID-19 and aged between 30 and 69 years; (2) having presented moderate to severe symptoms; (3) having received clinical discharge after at least 30 days (if they had been hospitalized) or it had been 30 days since the recovery of clinical signs. Exclusion criteria were: (1) being exposed in the last three months to altitude >1500 m; (2) having significant physical limitations to performing the assessments or the intervention; (3) having any acute or chronic illness without medical supervision; (4) diagnosis of anemia; (5) use of immunosuppressive drugs; (6) pregnancy; (7) hormone replacement; (8) excessive use of alcohol or drugs; (9) smoking; (10) absences of three consecutive days during the intervention; and (11) attendance of less than 75% of the total number of sessions. Participants were randomly allocated into four groups according to the training (T) and recovery (R) association with hypoxia (H) or normoxia (N): (a) TH:RH, (b) TN:RH, (c) TN:RN and, last, (d) the control group (without training). For the allocation of participants, sex, age, physical condition of the participant (incremental test score) and severity during illness (COVID-19) were taken into account. Both the participants and the research team in charge of the evaluation sessions were blinded in this study.

The present study was approved by the School of Physical Education and Sport of Ribeirao Preto—University of Sao Paulo—Ethics Committees (CAAE: 33783620.6.0000.5659).

### 2.2. Intervention Protocol

Three training groups executed the cycling training program around the normobaric hypoxia tents (Colorado Altitude Tents—Colorado, USA) using the individual system, including the face mask. Training sessions were performed with a frequency of three sessions per week and consisted of three parts (warm-up, main session and cool down) with a total duration of up to 50.5 min. Three to six sets of 5 min with 2.5 min of passive recovery were performed, with an intensity close to the upper limit of training zone 2 (90–110% of the anaerobic threshold 2—AT2). Cardiorespiratory data are available in the study in [26]. The number of sets increased the training load during the first four weeks. Two groups of participants were exposed to hypoxia (Hypoxia and Hypoxia Recovery) at an inspired fraction of oxygen (FiO_2_) of ~13.5%, simulating approximately 3000 m of altitude. The normoxia group intervened at a FiO_2_ corresponding to sea level in the Multisport Gymnasium (~20.9%). All sessions were monitored by acute responses to the training using heart rate (HR), peripheral oxyhemoglobin saturation (SpO2), rate of perceived exertion (RPE; [27]) and the Lake Louise Scale [28]. In addition, the internal training load was quantified at each session through training impulse (TRIMP). More information on these variables can be found in a previous study of our group [29]. This procedure monitored possible differences in the absolute internal load among the TH:RH, TN:RH and TN:RN groups.

### 2.3. Data Collection

#### 2.3.1. PAR-Q and Anamnesis

Participants answered the Physical Activity Readiness Questionnaire (PAR-Q), and those who responded positively to any questions signed the Terms of Responsibility. In addition, sociodemographic data, general and specific (COVID-19) health statuses related to lifestyle, information on comorbidities and drug treatments were assessed through the Anamnesis. Finally, answers about the symptoms associated with COVID-19 allowed us to evaluate the intensity of the disease’s manifestation.

For the severity of the disease, participants were divided into four groups, based on the National Institutes of Health (NIH) of the United States of America criteria for COVID-19 severity [30]: (a) mild: present when there are any COVID-19 symptoms, such as fever, cough, etc., but without respiratory distress or dyspnea; (b) moderate: when there is presence of any COVID-19 symptoms and respiratory distress or dyspnea; (c) severe: persons who present any COVID-19 symptoms and need hospitalization, but not intensive care; (d) critical, when there is presence of any COVID-19 symptoms and need hospitalization and intensive care.

#### 2.3.2. Anthropometric Measurements

The body mass index (BMI) was used to measure nutritional status recommended by WHO [31] and waist–hip circumferences were used to calculate the waist–hip ratio (W/H) [32].

#### 2.3.3. Questionnaires

Self-reported physical activity: The International Physical Activity Questionnaire (IPAQ)—short version, validated in Brazil [33], measured the usual level of physical activity, with questions about the time spent being physically active in the last seven days. These instructions were given to certify that the possible differences between the groups after the intervention in the variables of interest are related to the physical training intervention’s effect, not other environmental factors. Regarding the IPAQ study of validity and reliability in Brazil [33], the sample consisted of 257 men and women who answered the IPAQ (usual week version, short and long forms) at the beginning of the study and after seven days. Part of the sample (n = 28) used the Computer Science & Applications (CSA) movements sensor to validate the instrument. Questionnaire reliability was determined after seven days, and Spearman’s correlation was significant and high (rho = 0.69–0.71: *p* < 0.01). The CSA validity was 0.46 for the long form and 0.75 for the short form. The study showed that the long and short forms are comparable, and the usual week and the last seven days of reference periods present similar results. It was observed that the forms of the IPAQ were acceptable and showed similar results to other instruments to measure physical activity levels.

Health-related quality of life (QoL): The 12-Item Health Survey (SF-12v1) instrument, composed of 12 items with Likert-type and dichotomous responses, was used. This instrument is derived from the Short Form Questionnaire 36 (SF-36), which assigns four domains (physical, functional, emotional and social) and eight subscales (physical functioning, physical role, bodily pain, general health, vitality, social functioning, emotional role and mental health). These subscales are used to calculate the physical (PCS) and mental (MCS) components. This version was translated and validated for Portuguese [34]. For the validation of the SF-12 v1 in Brazil, a population-based study was carried out on 779 patients with chronic obstructive pulmonary disease (COPD) and control participants, testing the reliability and correlations with the SF-36, Airways questionnaire 20 (AQ20) and the Saint George Hospital in Respiratory Disease (SGRQ). In the validation study, both SF12 domains were tested. As a result, the physical component reached rho = 0.69, *p* < 0.012, and the mental component reached rho = 0.63, *p* < 0.005.

Mental Health Status: The Depression, Anxiety and Stress Scale (DASS-21) [35] was used and the calculation of the score was based on a previous study [36]. The DASS-21 is composed of seven Likert-type response items (from strongly disagree to strongly agree). The total score indicates having a risk of depression, anxiety and stress, and each subscale is classified into five levels: normal, mild, moderate, severe and extremely severe. Finally, participants were divided into no risk (normal) or at risk (mild, moderate, severe and extremely severe). The instrument has been adapted and validated in Portuguese, having a Cronbach’s alpha of 0.92 for depression, 0.90 for stress and 0.86 for anxiety, indicating good internal consistency for each subscale.

#### 2.3.4. Statistical Analysis

The information described is shown as follows: continuous variables were expressed in basic descriptive statistics, using summary measures, while categorical variables were described in frequency and percentage. The effects of different types of training (control, hypoxia, normoxia), evaluated over time (before and after eight weeks of intervention), were compared using a linear regression model with mixed effects (fixed and random), including BMI and severity of the disease as covariates. Possible associations of categorical variables at baseline were checked using Fisher’s exact test. The significance level adopted was 5% in all analyses, and the support software was SAS (9.2 version).

## 3. Results

### 3.1. Participant’s Characteristics

Forty participants were excluded after the initial visit since they had either exceeded 60 days of disease recovery or did not have a positive COVID-19 diagnosis. Of the 84 participants who performed the first evaluation, 67 completed the evaluation eight weeks after the intervention (Figure 1). No injuries or health problems were detected during the intervention in any of the study participants.

The sample’s sociodemographic and health status characteristics are presented in Table 1, according to the control or training groups. No statistically significant differences were found between the groups for these characterization variables. In the total sample, women represented 64.2%, and 82.1% were under 60 years of age. On the other hand, 17.9% have been diagnosed with diabetes, 22.4% with hypertension, 79.1% are overweight or obese, 53.7% have metabolic syndrome and 4.5% have asthma. Concerning the severity of COVID-19, 79.1% had from moderate to critical symptoms, 17.9% were hospitalized and of these, 41.7% needed a mechanical respirator. Hypoxia groups showed significantly lower values of peripheral oxyhemoglobin saturation (SpO2) compared with the control group (GH 87.7% (6.5) and GHR 95.1% (3.1)). More information on these variables can be found in Costa, et al. 2022 [29]. 

### 3.2. Quality of Life and Mental Health

The effects of the intervention on the QoL and mental health of adults who had recovered from COVID-19 are shown in Table 2. No differences were observed in the control group. All training groups show statistically significant improvements in the physical and mental components of QoL after the intervention compared to baseline values (physical: normoxia *p* = 0.001; hypoxia recovery *p* < 0.001; hypoxia *p* = 0.004, mental: normoxia *p* = 0.006; hypoxia recovery *p* = 0.001; hypoxia *p* = 0.005). In the comparison between groups, the Hypoxia Recovery group showed significant differences in the physical component compared with the control group. In the mental component, the normoxia group shown significant differences compared with the control group. The highest percentage of improvement in the physical (17.7%) and mental (14.1%) components was shown in the Hypoxia Recovery Group.

In mental health, a significant statistical improvement versus baseline was shown in all training groups to stress (normoxia *p* < 0.001; hypoxia recovery *p* = 0.011; hypoxia *p* = 0.009) and anxiety variables (normoxia *p* = 0.002; hypoxia recovery *p* = 0.037; hypoxia *p* = 0.006). Normoxia and hypoxia groups also showed significant improvements versus baseline in the depression variable (normoxia *p* < 0.001; hypoxia *p* = 0.038). In the comparison among groups, in the baseline, the Hypoxia group showed higher significant values compared with the Control (*p* = 0.032) and Hypoxia Recovery (*p* = 0.043) groups in the DASS-21 stress scores. These differences are maintained after intervention between Hypoxia and Hypoxia Recovery groups (*p* = 0.038). The Normoxia group showed a higher percentage of improvement in the depression (66.7%), anxiety (46.5%) and stress (40.6%) components.

In relation to the risk of depression, anxiety and stress (Table 3), there was no statistical evidence in the comparison of the risk before and after the intervention; however, there was a decrease in the clinical relevance in each of the training sessions. The Normoxia group decreased the risk of depression by 28.6%, 26.7% in the Hypoxia Recovery group and 17.7% in the Hypoxia group. The risk of anxiety decreased by 21.4% in the Normoxia group, 26.7% in the Hypoxia Recovery group and 17.7% in the Hypoxia group. Finally, the risk of stress decreased by 20% in the Hypoxia Recovery group and 5.9% in the Hypoxia group. There was no decrease in the Normoxia group.

## 4. Discussion

The current study investigated the effects of moderate-intensity intermittent hypoxic training in adults who had recovered from COVID-19 on QoL and the state of mental health through the assessment of levels of depression, anxiety and stress. The main results of this study show that moderate-intensity normoxic/hypoxic intermittent training improves the health-related QoL and the mental health. However, the results did not show statistical changes when comparing the risk of depression, anxiety and stress before and after each intervention.

Different studies have analyzed the presence of persistent symptoms or sequelae after recovery from COVID-19 [17,37]. For example, a study conducted in Italy indicates that the symptoms of the disease persist 60 days after the onset, affecting their health-related QoL [38]. In this context, a pre-pandemic systematic review indicated that it takes up to one year for people who have recovered from a serious illness to regain their QoL. The greatest effectiveness of health interventions is shown during the first year after discharge from the hospital [39]. However, few researchers still analyze the short-, medium- and long-term effects of COVID-19 on the patient’s general health, physical function and reintegration into the workplace post-infection [40,41]. In this context, it is necessary to contribute updated and contextualized evidence to the Brazilian reality.

The present study consisted of 67 adults who recovered from COVID-19, of whom 17.9% required hospitalisation, and of these, 41.7% required mechanical ventilation. After the moderate-intensity intermittent normoxic/hypoxic training intervention, an increase of 15.7% was shown in the Normoxia group, 17.7% in Hypoxia Recovery and 11.6% in Hypoxia in the physical component of QoL. In the mental dimension, the findings showed an increase of 11.2% in the Normoxia group, 14.1% in Hypoxia Recovery and 11.8% in Hypoxia. These results were expected as there is previous evidence that hypoxic training can positively affect health-related QoL [13,18,20,21,22].

Mental health results from DASS-21 also show an effect of moderate-intensity intermittent normoxic/hypoxic training on depressive symptoms (66.7% reduction in the normoxia group, 31.2% reduction in the hypoxia recovery group and 31% in the hypoxia group), anxiety symptoms (46.5% decrease in the Normoxia group, 45.9% in the Hypoxia Recovery group and 39.5% in the Hypoxia group) and stress symptoms (40.6% decrease in the Normoxia group, 36.3% in the Hypoxia Recovery group and 22.1% in the Hypoxia group). The effects of the three training groups show statistically significant differences between the baseline and post-intervention measurements. However, there were no statistical changes when comparing the risk of depression, anxiety and stress before and after each intervention. Nonetheless, it is important to highlight a decrease in clinical relevance in the groups that perform physical exercise due to the risk of depression (of the order of 28.6% in the Normoxia group, 27.7% in the Recovery Hypoxia Recovery group and 17.7% in the Hypoxia group) and anxiety (21.4% in the Normoxia group, 26.7% in the Hypoxia Recovery group and 17.6% in the Hypoxia group). These results are relevant, considering that it is estimated that the prevalence of mental disorders during the pandemic will increase by more than 20% [42]. For stress, only the Hypoxia Recovery group showed a 20% decrease. Although severe chronic hypoxia may promote anxiety disorders and depression, a more moderate dose could increase cellular capacity and psychological resilience, considering a new treatment strategy for mental health [19].

Although unlike previous studies, no statistically significant differences were established with the normoxia group, and the main advantage of our interventions in hypoxia is the possible reduction in mechanical stress that these groups can develop compared to in normoxia. Previous studies have shown how exercise in hypoxic situations decreases the external load while the internal load remains constant [43]. In previous work by our group (Armored), where monitoring values of the present interventions are shown, parameters such as rate of perceived exertion (RPE), lactate concentration, relative heart rate (HR) or other internal load parameters (as training impulse) did not show statistically significant differences between groups. However, compared to normoxia, oxygen saturation (SpO2) differences were shown between hypoxia groups. Although external load parameters were not evaluated in the previous work, other authors have established how there is a 2% reduction in VO_2_MAX for every 1% reduction in SpO2 [44,45]. Thus, it can be assumed that there was a reduction in the external load, mainly in the hypoxia group. Therefore, after obtaining improvements in the QoL and mental health in all intervention groups, exposure to hypoxia can benefit people with special health conditions because the ergogenic effects would be achieved with less mechanical stress [46]. Although the cause–effect relationship has not been studied, the improvements in these parameters could be related to the increases observed in our previous study on the cardiovascular system [26]. Our previous findings establish how moderate-intensity training in normoxia or hypoxia promoted similar benefits in the cardiovascular fitness of this population. In addition, hypoxia offered an additional stimulus to training by promoting an increase in erythropoietin and hematological stimulation. Therefore, it is suggested that public policies can offer care to people who have recovered from COVID-19 and that hypoxic training can be considered a gradual recovery process that these people face in biological and psychological aspects. Furthermore, due to their disability, these patients can suffer after recovering from the disease; the hypoxic stimulus could be a good strategy for those individuals who cannot be subjected to great effort.

Some limitations should also be mentioned. The most important limitation is that the sample size of the present study is relatively small, due to the difficulty of access to the participants at the time of the intervention, with Brazil being under mobility restrictions with high infection levels and mortality. So, achieving this number of participants to develop the study was very important.

Complementary information related to the study design can be found in the study protocol [24]. However, it is important to highlight an innovation. In general, only one person could train and breathe the air from the tent. The system, which was developed with individual hoses and masks, allowed up to eight people to train simultaneously in the same tent, reaching a greater number of community participants. Furthermore, the cover that did not allow the participant to know whether the hose connected to their mask was inside or on the side of the tent, made blinding possible.

The development of this study in that pandemic moment, without vaccines available, was very important, and the cause–effect relationship between external mechanical stress and hypoxic stimulus could provide interesting findings on the benefits of hypoxia in a pathological population. The complementary points that enrich and support this work are the effectiveness [29] of an innovative protocol that uses hypoxia as an additional stimulus to the training and that could be applicable in other special health conditions.

## 5. Conclusions

Normoxic and hyoxic training showed similar effects on QoL’s physical and mental components and the mental health. Furthermore, in relation to the effectiveness of risk in each dimension, all types of training had clinical relevance. Therefore, based on the findings, the different modes of hypoxic training did not show additional effects compared with normoxic training.

## Figures and Tables

**Figure 1 healthcare-11-03076-f001:**
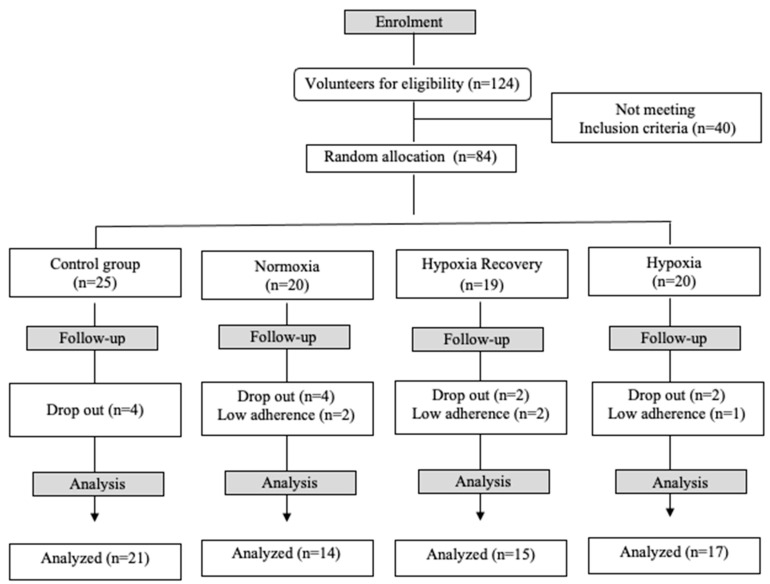
Flowchart of the study.

**Table 1 healthcare-11-03076-t001:** Descriptive characteristics of the participants in the baseline.

	Control (n = 21)	Normoxia (n = 14)	Hypoxia Recovery (n = 15)	Hypoxia (n = 17)	*p*
Age (years)	48.0 (10.7)	50.4 (9.5)	47.5 (9.9)	48.1 (9.9)	0.8
BMI (kg/m^2^)	29.6 (4.6)	29.0 (4.2)	29.2 (5.8)	30.7 (3.9)	0.6
W/H ratio	0.87 (0.09)	0.90 (0.06)	0.90 (0.10)	0.92 (0.10)	0.7
Total PA/week (min)	522.1 (464.3)	405.4 (511.5)	385.7 (376.8)	390.3 (469.4)	0.7
Total seated time/week (min)	2351.9 (1304.2)	2777.0 (928)	2234.0 (1023.1)	2735.9 (833.5)	0.3
Sex ^a^					
Male	8 (38.1)	3 (21.4)	6 (40.0)	7 (41.2)	0.6
Female	13 (61.9)	11 (78.6)	9 (60.0)	10 (58.8)	
Presence of diseases ^a^					
Diabetes	4 (19.1)	2 (14.3)	2 (13.3)	4 (23.5)	0.9
Hypertension	3 (14.3)	5 (35.7)	4 (26.7)	3 (17.7)	0.4
Asthma	0 (0.0)	1 (7.1)	1 (6.7)	1 (5.9)	0.6
COVID-19 Severity ^a^					0.5
Mild	5 (23.8)	3 (21.4)	3 (20.0)	3 (17.7)	
Moderate	11 (52.4)	7 (50.0)	10 (66.7)	13 (76.5)	
Severe	4 (19.1)	2 (14.3)	0 (0.0)	1 (5.9)	
Critical	1 (4.8)	2 (14.3)	2 (13.3)	0 (0.0)	

Values are given as the mean (standard deviation) *p*-values of the ANOVA. ^a^ Values expressed as n (%) *p*-values of the Fisher test. BMI: body mass index; W/H: waist–hip ratio.

**Table 2 healthcare-11-03076-t002:** Effects of normoxia and hypoxia training interventions on quality of life and mental health of patients who had recovered from COVID-19.

	Control Group (n = 21)			Normoxia (n = 14)			Hypoxia Recovery (n = 15)			Hypoxia (n = 17)		
	Baseline	Post	Δ%	Baseline	Post	Δ%	Baseline	Post	Δ%	Baseline	Post	Δ%
SF-12 Physical score	47.3 (7.9)	47.1 (8.3)	−0.4	42.1 (11.0)	48.7 (7.0) *	15.7	45.2 (7.7)	53.2 (3.8) *^$^	17.7	45.8 (9.2)	51.1 (5.3) *	11.6
SF-12 Mental score	47.4 (10.8)	46.9 (11.3)	−1.1	48.8 (7.9)	54.6 (4.6) *^$^	11.2	46.9 (11.8)	53.5 (6.6) *	14.1	46.5 (9.7)	52.0 (9.9) *	11.8
DASS-21 Depression score	7.9 (8.8)	7.6 (7.7)	−3.8	9.3 (9.1)	3.1 (3.8) *	−66.7	6.4 (6.1)	4.4 (5.6)	−31.2	10.0 (10.4)	6.9 (7.9) *	−31.0
DASS-21 Anxiety score	7.1 (5.1)	6.8 (6.7)	−4.2	11.4 (11.7)	6.1 (6.8) *	−46.5	7.2 (7.2)	3.9 (3.9) *	−45.9	10.9 (9.6)	6.6 (5.7) *	−39.5
DASS-21 Stress score	11.5 (9.4) ^α^	12.3 (9.6)	6.9	15.3 (10.3)	9.1 (6.7) *	−40.6	11.3 (8.7) ^α^	7.2 (5.7) *^α^	−36.3	18.1 (10.3)	14.1 (9.4) *	−22.1

Values expressed as n (%). SF-12: (Short-Form Health Survey); DASS: (Depression, Anxiety and Stress Scale); Significant differences: * Significant differences vs. Baseline *p* < 0.05; ^$^ Significant differences vs. Control Group *p* < 0.05; ^α^ Significant differences vs. Hypoxia Group *p* < 0.05.

**Table 3 healthcare-11-03076-t003:** Comparison of participants’ risk of depression, anxiety and stress before and after the intervention.

	Depression			Anxiety			Stress		
	No Risk	Risk	*p*	No Risk	Risk	*p*	No Risk	Risk	*p*
Control Group									
Baseline (%)	13 (61.9)	8 (38.1)	0.999	10 (47.6)	11 (52.4)	0.758	10 (47.6)	11 (52.4)	0.999
Post (%)	12 (57.1)	9 (42.9)		12 (57.1)	9 (42.9)		10 (47.6)	11 (52.4)	
Normoxia									
Baseline (%)	9 (64.3)	5 (35.7)	0.165	6 (42.9)	8 (57.1)	0.450	8 (57.1)	6 (42.9)	0.999
Post (%)	13 (92.9)	1 (7.1)		9 (64.3)	5 (35.7)		8 (57.1)	6 (42.9)	
Hypoxia Recovery									
Baseline (%)	9 (60.0)	6 (40.0)	0.215	9 (60.0)	6 (40.0)	0.215	7 (46.7)	8 (53.3)	0.462
Post (%)	13 (86.7)	2 (13.3)		13 (86.7)	2 (13.3)		10 (66.7)	5 (33.3)	
Hypoxia									
Baseline (%)	10 (58.8)	7 (41.2)	0.465	7 (41.2)	10 (58.8)	0.494	5 (29.4)	12 (70.6)	0.999
Post (%)	13 (76.5)	4 (23.5)		10 (58.8)	7 (41.2)		6 (35.3)	11 (64.7)	

Values expressed as n (%). *p*-Values of the Fisher Exact Test.

## Data Availability

Data are available upon request.

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
