# Peer review of "Effects of Different Protocols of Moderate-Intensity Intermittent Hypoxic Training on Mental Health and Quality of Life in Brazilian Adults Recovered from COVID-19: The AEROBICOVID Double-Blind Randomized Controlled Study"

_healthcare, 2023, doi:10.3390/healthcare11233076_

Round 1
Reviewer 1 Report
Comments and Suggestions for Authors
Dear authors,
The structure and development of the paper are well-defined and well-thought-out. The theoretical foundation is pertinent, and the methodological development is appropriate for the type of study.
Before publication, in my opinion, some minor revisions are necessary, as listed below:
Methods - Authors must specify the exact period of time in which the databases were collected and the Intervention Protocol was carried out.
Indicate the limitations of the study in the last paragraph of the discussion section. I suggest inserting two paragraphs before the conclusions, one with the limitations of the study and another one with the strong points of the research
Considering the findings of the study, the authors should improve the conclusions section.
The authors may also consider clarifying the title, including in the case of which country. This point is of discretions of them. However, it will make the title more relevant to the content of the article.
Best regards!
Author Response
Comment 1: Methods - Authors must specify the exact period of time in which the databases
were collected and the Intervention Protocol was carried out.
Answer 1. The protocol was carried out from September to December 2020.
Comment 2: Indicate the limitations of the study in the last paragraph of the discussion section. I
suggest inserting two paragraphs before the conclusions, one with the limitations of the study and
another one with the strong points of the research
Answer 2. Thanks for your suggestions. The discussion section has been completed with
limitations and strengths:
Some limitations should also be mentioned. The most important limitation is that the sample size
of the present study is relatively small, due to the difficulty of access to the participants at the
time of the intervention, Brazil being under mobility restrictions with high infection levels and
mortality. On the other hand, the development of this study in that pandemic moment, without
vaccines available was so important, and the cause-effect relationship between external
mechanical stress and hypoxic stimulus could provide interesting findings on the benefits of
hypoxia in a pathological population. The complimentary points that enrich and support this
work are the effectiveness of an innovative protocol that uses hypoxia as an additional stimulus
to the additional training and that is applicable to people who are inserted in the community.
Comment 3: Considering the findings of the study, the authors should improve the conclusions
section.
Answer 3. Conclusions have been reviewed. Effectively, the main finding of this study is that both
normoxic and hypoxic protocol achieve improvements in QoL and mental health in the
participants. This should be the conclusion of our work. Thank you
Comment 4: The authors may also consider clarifying the title, including in the case of which
country. This point is of discretions of them. However, it will make the title more relevant to the
content of the article.
Answer 4. Authors appreciate the review of the reviewer. The title has been completed.

Reviewer 2 Report
Comments and Suggestions for Authors
To the authors
The study entitled “Effect of moderate-intensity intermittent hypoxic training on mental health and quality of life in adults recovered from COVID-19: the AEROBICOVID double-blind randomized controlled study” presents the effects of an 8-week aerobic cycle exercise training on mental health and quality of life in recovered patients from COVID-19.
The authors trained patients on three different conditions: normoxia (training and recovery), hypoxia (training and recovery), and hypoxia recovery (training in normoxia and recovery in hypoxia). Mental Health and quality of life variables were compared among training groups after the training period. All training groups demonstrated differences between the control group (not trained participants).
The study was properly conducted based on a good methodology. However, I think that the mental health and quality of life outcomes were not the main aims of the study. Some major revisions need to be done in order to make the messages conveyed from the study to be interesting. I list some below.
Title
I think that the title does not accurately represent the study's results.
Presenting that the moderate-intensity intermittent hypoxic training was the only one that affected the mental health and the quality of life of adults recovered from COVID-19 is biased, as all the training groups presented similar results at the end of the 8-week training.
It would be more representative if the title was like this:
“Effects of hypoxic and normoxic training on …. study”
Abstract
In this section should be presented the main results of the study, adding values of the pre and post-training outcomes, (e.g. QoL pre and post-training in each group), instead of repeating the same information.
Specify in the abstract the groups as they are presented in the section on the methods (lines 135-136) for a better understanding
I suggest deleting the sentences: “However, despite the clinical decrease in symptoms,……and QoL to the normoxia group, thus helping recovery” as these sentences do not add any important information rather than repeating the aforementioned findings.
“Exposure to hypoxic training, subjecting participants to less mechanical load, would present similar benefits on mental health and QoL to the normoxia group, thus helping the recovery”. This sentence is confusing and it does not stand properly here.
Introduction
This section is too long and it does not give a good rationale for the study, based on the effects of training on the improvement of the physiology of mental health and therefore to the improvement of the QoL. Indeed exercise improves physical capacity, cognitive performance and indirectly QoL. I think that the authors should give a better rationale for the use of different exercise training programs (e.g. normoxia, hypoxia) that may differentiate the level of the expected improvement of the mental and physical outcomes.
Line 68: Number 11 reference, is not relevant to the statement “Different Exercise Protocols …adults from COVID-19”. Please find another one.
I suggest authors to remove studies on animals.
Materials and Methods
Please explain the age limitation of 30-69 years. Why 69 years instead of 70 years ?
Line 136: In this point I suggest authors to write how the control group was treated? Did it participate in any kind of exercise or not?
Lines 138-140: “Blinding was done ….and the participants”.
What does the reference number 25 add to the blinding process? I think that you do not need to add a reference in this sentence.
The IPAQ for reporting physical activity level is not needed to be included in your study as you have not presented any results and the use of the IPAQ and this questionnaire does not add any important information.
Results
In Table 1 merge the presence or not of diabetes, hypertension, asthma in one line for each pathology, instead of having 2 lines (e.g. diabetes yes / diabetes no)
The label of Table 2 is biased.
Replace the phrase “Effects of moderate-intensity …..on quality of life….” with the phrase: “Effects of normoxia and hypoxia training interventions on quality of life….”. At the bottom of the Table 2, add p values for each significant difference, e.g. *significant differences vs baseline p<??? ;
Table 3 is not mentioned in the section of the results.
Why do you present the risk of depression/anxiety and stress before and after the intervention? This aim was not mentioned in the study. How did you calculate the risk?
I am not sure that this is clear and that it adds value to your study’s results. So I suggest you remove this table from the manuscript.
Discussion
Although there are important information in this section of the manuscript, the cause - effect relationship (less external mechanical stress in hypoxia training – improvements in the QoL and mental health) in not strong explained, resulting to a weak discussion. To strengthen your discussion, I think that you have to focus on
the effects of all types of exercise training
Line 367: clarify the abbreviations TRIMPHR, TRIMPRPE
Author Response
Reviewer 2
The study entitled “Effect of moderate-intensity intermittent hypoxic training on mental health and quality of life in adults recovered from COVID-19: the AEROBICOVID double-blind randomized controlled study” presents the effects of an 8-week aerobic cycle exercise training on mental health and quality of life in recovered patients from COVID-19.
The authors trained patients on three different conditions: normoxia (training and recovery), hypoxia (training and recovery), and hypoxia recovery (training in normoxia and recovery in hypoxia). Mental Health and quality of life variables were compared among training groups after the training period. All training groups demonstrated differences between the control group (not trained participants).
The study was properly conducted based on a good methodology. However, I think that the mental health and quality of life outcomes were not the main aims of the study. Some major revisions need to be done in order to make the messages conveyed from the study to be interesting. I list some below.
Comment 1- Title - I think that the title does not accurately represent the study's results. Presenting that the moderate-intensity intermittent hypoxic training was the only one that affected the mental health and the quality of life of adults recovered from COVID-19 is biased, as all the training groups presented similar results at the end of the 8-week training. It would be more representative if the title was like this: “Effects of hypoxic and normoxic training on …. study”
Answer 1. Firstly, the authors are grateful for the thorough revision of the manuscript. The title has been revised and completed. In relation to the inclusion of training in normoxia, although they also improve your parameters, as we have detailed in the conclusions, we believe that this is not the novelty of the research. The novelty is to study the synergistic effects of exercise and hypoxia (with different hypoxia protocols) on these parameters. This is what we have already detailed in the title. Thanks again.
Comment 2- Abstract - In this section should be presented the main results of the study, adding values of the pre- and post-training outcomes, (e.g. QoL pre and post-training in each group), instead of repeating the same information. Specify in the abstract the groups as they are presented in the section on the methods (lines 135-136) for a better understanding
Answer 2. Abstract have been completed with the methodological information. Results have also been presented: “The sample of this clinical trial-controlled double-blind study consisted of 67 participants aged 30-69 years, whose were randomly according to Normoxia, Hypoxia, Hypoxia Recovery or Control Group. Eight weeks of cycle ergometer training were performed with a frequency of 3 training sessions per week in normoxic or hypoxic conditions (with or without hypoxic recovery). Health-related QoL and Mental Health Status were evaluated by 12-Item Short Form Survey and Depression Anxiety and Stress Scale instruments, respectively. All training groups improved the QoL's physical (Baseline – Post: Normoxia Group 42.1 (11.0) - 48.7 (7.0), Hypoxia Group 46.9 (11.8) - 53.5 (6.6) and Hypoxia Recovery Group 45.8 (9.2) - 51.1 (5.3)) and mental (Baseline – Post: Normoxia Group 48.8 (7.9) - 54.6 (4.6), Hypoxia Group 45.2 (7.7) - 53.2 (3.8) and Hypoxia Recov-ery Group 46.5 (9.7) - 52.0 (9.9)) dimensions. Regarding mental health outcomes, all training groups decreased depressive symptoms (66.7% Normoxia, 31.2% Hypoxia Recovery, and 31% Hypoxia groups), anxiety symptoms (46.5% Normoxia, 45.9% Hypoxia Recovery and 39.5% in the Hypoxia groups) and stress symptoms (40.6% Normoxia, 36.3% Hypoxia Recovery and 22.1% Hypoxia groups). Significant statistical difference was not found between groups”.
Comment 3 - Abstract - I suggest deleting the sentences: “However, despite the clinical decrease in symptoms,……and QoL to the normoxia group, thus helping recovery” as these sentences do not add any important information rather than repeating the aforementioned findings.“Exposure to hypoxic training, subjecting participants to less mechanical load, would present similar benefits on mental health and QoL to the normoxia group, thus helping the recovery”. This sentence is confusing and it does not stand properly here.
Answer 3. As indicated in the previous comment, the abstract has been improved. Thank you again.
Comment 4- Introduction -This section is too long and it does not give a good rationale for the study, based on the effects of training on the improvement of the physiology of mental health and therefore to the improvement of the QoL. Indeed exercise improves physical capacity, cognitive performance and indirectly QoL. I think that the authors should give a better rationale for the use of different exercise training programs (e.g. normoxia, hypoxia) that may differentiate the level of the expected improvement of the mental and physical outcomes.
Answer 4. The Introduction has been revised, thanks for your suggestions.
Comment 5- Introduction - Line 68: Number 11 reference, is not relevant to the statement “Different Exercise Protocols …adults from COVID-19”. Please find another one.
Answer 5. The reference have been replaced by Amini, A., Vaezmousavi, M., & Shirvani, H. (2023). Comparing the effect of individual and group cognitive-motor training on reconstructing subjective well-being and quality of life in older males, recovered from the COVID-19. Cognitive Processing, 1-14. Thanks.
Comment 6 - Introduction - I suggest authors to remove studies on animals.
Answer 6. Although we appreciate the reviewer's contribution, we believe that this type of study is useful for a better understanding of the effects of different treatments, especially in those topics where the scientific literature is not very extensive, as is the case here. The introduction has been revised so that it is neither confusing nor lengthy. We hope that the result is now satisfactory.
Comment 7- Introduction Materials and Methods - Please explain the age limitation of 30-69 years. Why 69 years instead of 70 years?
Answer 7. As mentioned in lines 176-177, participants answered the Physical Activity Readiness Questionnaire (PAR-Q). In this questionnaire, participants answer questions about their health and sign a responsibility term. The law of Sao Paulo state in Brazil allows people until 69 years old to use this questionnaire. From 70 years old the law indicates a document from a physician to practice physical activity.
Comment 8- Introduction Materials and Methods - Line 136: In this point I suggest authors to write how the control group was treated? Did it participate in any kind of exercise or not?
Answer 8. The information was added. The control group did not participate in any kind of exercise.
Comment 9- Introduction Materials and Methods - Lines 138-140: “Blinding was done ….and the participants”.
Answer 9. It is true that this sentence is confusing. It has been revised: “Both, the participants and the research team in charge of the evaluation sessions were blinded in this study”
Comment 10- Introduction Materials and Methods - What does the reference number 25 add to the blinding process? I think that you do not need to add a reference in this sentence.
Answer 10. Effectively, there are a mistake. The reference was removed.
Comment 11- Introduction Materials and Methods - The IPAQ for reporting physical activity level is not needed to be included in your study as you have not presented any results and the use of the IPAQ and this questionnaire does not add any important information.
Answer 11: The research group decided to include data to present the pre-intervention characteristics of each group. In addition, the level of physical activity was used as a variable control, to ensure that participants did not participate in another physical exercise programme during the intervention.
Comment 12- Results -In Table 1 merge the presence or not of diabetes, hypertension, asthma in one line for each pathology, instead of having 2 lines (e.g. diabetes yes / diabetes no)
Answer 12: The table have been corrected.
Comment 13- Results - The label of Table 2 is biased.
Answer 13: The label is completed.
Comment 14- Results - Replace the phrase “Effects of moderate-intensity …..on quality of life….” with the phrase: “Effects of normoxia and hypoxia training interventions on quality of life….”.
Answer 14: The correction has been made.
Comment 15- Results -At the bottom of the Table 2, add p values for each significant difference, e.g. *significant differences vs baseline p<???
Answer 15: The information was added, thanks.
Comment 16- Results -Table 3 is not mentioned in the section of the results.
Answer 16: This information has been added. Sorry for the inconvenience
Comment 17- Results -Why do you present the risk of depression/anxiety and stress before and after the intervention? This aim was not mentioned in the study. How did you calculate the risk? I am not sure that this is clear and that it adds value to your study’s results. So I suggest you remove this table from the manuscript.
Comment 18- Discussion -Although there are important information in this section of the manuscript, the cause - effect relationship (less external mechanical stress in hypoxia training – improvements in the QoL and mental health) in not strong explained, resulting to a weak discussion. To strengthen your discussion, I think that you have to focus on the effects of all types of exercise training
Answer 18: Indeed, the authors agree that this cause-effect relationship has not been evaluated in this paper. These statements are based on the results published in a previous article, belonging to the same research project. Thus, we have revised the conclusions of this paper. The above point is mentioned only in the discussion, in order to know the benefits of using these new training strategies. Limitations have been added in this sense. Thank you.
Comment 19- Discussion Line 367: clarify the abbreviations TRIMPHR, TRIMPRPE
Answer 19: This abbreviature have been replace by your means, training impulse, as parameter of internal load. Thanks.

Reviewer 3 Report
Comments and Suggestions for Authors
I appreciate the topical and still undiscovered research topic. The introduction to the article is sufficient, the description of the methodology is exhaustive, and the results are presented correctly, although I would like to ask you to change the presentation of Figure 1 - it is a scan, which will make it difficult for the reader to understand.
I am satisfied with the discussion, but the conclusions should be extended, especially to include research limitations and further research directions, especially due to the importance of the topic.
Author Response
Reviewer 3
I appreciate the topical and still undiscovered research topic. The introduction to the article is sufficient, the description of the methodology is exhaustive, and the results are presented correctly, although
Comment 1- I would like to ask you to change the presentation of Figure 1 - it is a scan, which will make it difficult for the reader to understand.
Answer 1: We have attached a new version of the Figure 1. Apologies for the inconvenience
Comment 2- I am satisfied with the discussion, but the conclusions should be extended, especially to include research limitations and further research directions, especially due to the importance of the topic.
Answer 2: We have reviewed and rewrite the conclusions of this paper. Limitations and strength have been added. Thank you by your review.

Round 2
Reviewer 2 Report
Comments and Suggestions for Authors
the revision of your manuscript is accepted
Author Response
The authors of this work are grateful for the efforts of the reviewer.
His contributions have been of great relevance to us. We consider that the manuscript has been improved by his suggestions.
Thank you again for your involvement
Reviewer 3 Report
Comments and Suggestions for Authors
thank you for the revised version